# Characterization of Epithelial–Mesenchymal and Neuroendocrine Differentiation States in Pancreatic and Small Cell Ovarian Tumor Cells and Their Modulation by TGF-β1 and BMP-7

**DOI:** 10.3390/cells13232010

**Published:** 2024-12-05

**Authors:** Hendrik Ungefroren, Juliane von der Ohe, Rüdiger Braun, Yola Gätje, Olha Lapshyna, Jörg Schrader, Hendrik Lehnert, Jens-Uwe Marquardt, Björn Konukiewitz, Ralf Hass

**Affiliations:** 1First Department of Medicine, University Hospital Schleswig-Holstein, Campus Lübeck, 23538 Lübeck, Germany; 2Institute of Pathology, University Hospital Schleswig-Holstein, Campus Kiel, 24105 Kiel, Germany; 3Biochemistry and Tumor Biology Lab, Department of Obstetrics and Gynecology, Hannover Medical School, 30625 Hannover, Germany; 4Department of Surgery, University Hospital Schleswig-Holstein, Campus Lübeck, 23538 Lübeck, Germany; 5First Department of Medicine, University Hospital Hamburg-Eppendorf, 20251 Hamburg, Germany; 6Warwickshire Institute for the Study of Diabetes, Endocrinology and Metabolism (WISDEM), University Hospitals Coventry and Warwickshire (UHCW), Coventry CV2 2DX, UK

**Keywords:** pancreatic ductal adenocarcinoma, epithelial–mesenchymal transition, mesenchymal–epithelial transition, neuroendocrine differentiation, TGF-β

## Abstract

Pancreatic ductal adenocarcinoma (PDAC) has an extremely poor prognosis, due in part to early invasion and metastasis, which in turn involves epithelial–mesenchymal transition (EMT) of the cancer cells. Prompted by the discovery that two PDAC cell lines of the quasi-mesenchymal subtype (PANC-1, MIA PaCa-2) exhibit neuroendocrine differentiation (NED), we asked whether NED is associated with EMT. Using real-time PCR and immunoblotting, we initially verified endogenous expressions of various NED markers, i.e., chromogranin A (CHGA), synaptophysin (SYP), somatostatin receptor 2 (SSTR2), and SSTR5 in PANC-1 and MIA PaCa-2 cells. By means of immunohistochemistry, the expressions of CHGA, SYP, SSTR2, and the EMT markers cytokeratin 7 (CK7) and vimentin could be allocated to the neoplastic ductal epithelial cells of pancreatic ducts in surgically resected tissues from patients with PDAC. In HPDE6c7 normal pancreatic duct epithelial cells and in epithelial subtype BxPC-3 PDAC cells, the expression of CHGA, SYP, and neuron-specific enolase 2 (NSE) was either undetectable or much lower than in PANC-1 and MIA PaCa-2 cells. Parental cultures of PANC-1 cells exhibit EM plasticity (EMP) and harbor clonal subpopulations with both M- and E-phenotypes. Of note, M-type clones were found to display more pronounced NED than E-type clones. Inducing EMT in parental cultures of PANC-1 cells by treatment with transforming growth factor-β1 (TGF-β1) repressed epithelial genes and co-induced mesenchymal and NED genes, except for SSTR5. Surprisingly, treatment with bone morphogenetic protein (BMP)-7 differentially affected gene expressions in PANC-1, MIA PaCa-2, BxPC-3, and HPDE cells. It synergized with TGF-β1 in the induction of vimentin, SNAIL, SSTR2, and NSE but antagonized it in the regulation of CHGA and SSTR5. Phospho-immunoblotting in M- and E-type PANC-1 clones revealed that both TGF-β1 and, surprisingly, also BMP-7 activated SMAD2 and SMAD3 and that in M- but not E-type clones BMP-7 was able to dramatically enhance the activation of SMAD3. From these data, we conclude that in EMT of PDAC cells mesenchymal and NED markers are co-regulated, and that mesenchymal–epithelial transition (MET) is associated with a loss of both the mesenchymal and NED phenotypes. Analyzing NED in another tumor type, small cell carcinoma of the ovary hypercalcemic type (SCCOHT), revealed that two model cell lines of this disease (SCCOHT-1, BIN-67) do express *CDH1*, *SNAI1*, *VIM*, *CHGA*, *SYP*, *ENO*2, and *SSTR2*, but that in contrast to BMP-7, none of these genes was transcriptionally regulated by TGF-β1. Likewise, in BIN-67 cells, BMP-7 was able to reduce proliferation, while in SCCOHT-1 cells this occurred only upon combined treatment with TGF-β and BMP-7. We conclude that in PDAC-derived tumor cells, NED is closely linked to EMT and TGF-β signaling, which may have implications for the therapeutic use of TGF-β inhibitors in PDAC management.

## 1. Introduction

Pancreatic cancer is one of the deadliest cancers worldwide. It comprises two major histological subtypes: pancreatic ductal adenocarcinoma (PDAC), which accounts for 90% of all cases, and pancreatic neuroendocrine neoplasm (panNEN), making up 3–5% of all cases. PanNEN is further subgrouped into well-differentiated pancreatic NE tumors and poorly differentiated pancreatic NE carcinoma (panNEC). PDAC and panNEN are considered different diseases with distinct biology, cells of origin, and genomic alterations, however, it is conceivable that PDAC and panNEC share common cells of origin. This is supported by molecular profiling data suggesting that panNEC is related to PDAC with respect to genetics and phenotype [1]. PanNENs represent a heterogeneous group of epithelial tumors with NE differentiation (NED) that are classified into well-differentiated pancreatic NE tumors (panNETs), including G1, G2, and G3 tumors, and poorly differentiated panNECs [2]. PanNETs can be regarded as a unique category, where G1-G2 tumors may progress to G3 tumors mainly driven by DAXX/ATRX mutations [2]. Conversely, panNECs display histomolecular features more closely related to PDAC, including TP53 and Rb alterations [2]. It was therefore not surprising that two PDAC-derived tumor cell lines of the quasi-mesenchymal subtype, PANC-1 and MIA PaCa-2, were demonstrated to harbor a NED phenotype [3]. PANC-1 cells express vimentin (VIM), CK5.6, MNF-116, chromogranin A (CHGA), neural cell adhesion molecule (NCAM/CD56), and somatostatin receptor-2 (SSTR2) but not E-cadherin (ECAD), synaptophysin (SYP), or neurotrophin receptor-1 (NTR1), while MIA PaCa-2 cells express VIM, CK5.6, AE1/AE3, CHGA, SSTR2, SYP, ECAD, and NTR1 but not NCAM [3]. In addition to the NED markers, we [4] and others [5] demonstrated that PANC-1 and MIA PaCa-2 cells express genes associated with endocrine/neuroendocrine differentiation such as *MAFA*, *NEURODI*, *PDX1*, and *NEUROG3*. Of note, PANC-1 cells exhibit epithelial–mesenchymal plasticity (EMP). Parental cultures comprise several clonal subpopulations with different EM transdifferentiation (EMT) phenotypes, of which some are more epithelial and others more mesenchymal in nature [6]. The importance of EMT is also considered in other gastrointestinal (GI) cancers such as colorectal NEC (coloNEC) and a distinct subtype of ovarian cancer (OC), e.g., the heterogeneous small cell carcinoma of the ovary hypercalcemic type (SCCOHT). The SCCOHT is a rare but aggressive type of ovarian cancer that is predominantly observed in young females. This malignant neoplasia is associated with paraneoplastic hypercalcemia, and affected patients often have a lethal outcome already within a few months after diagnosis [7,8]. Cellular models for this tumor entity were established and characterized as the SCCOHT-1 [9] and BIN-67 cell lines [10,11]. Indeed, the characterization of SCCOHT-1 cells and chemotherapeutic responses has revealed a constitutive expression of NED markers such as NCAM in the original patient tumor and derived mouse xenograft tumors [9,12]. Thus, the heterogeneity and plasticity of SCCOHT-1 cells may also involve EMP and maturation along a NED phenotype [13].

A series of studies suggests that NED is closely associated with EMT in different tumor types, mainly prostate, lung, and colon/pancreas. In prostate carcinoma (PCa), androgen deprivation activates both EMT and NE transdifferentiation programs [14]. Various factors have been implicated in the onset and progression of NED in prostate adenocarcinomas, such as androgen receptor (AR) loss, conventional therapy, and cytokine dysregulation. The AR is a crucial promoter of tumor progression and therapeutic response in patients with metastatic castrate-resistant PCa (CRPC) [15]. The acquisition of EMT and cancer stem cell characteristics may be closely linked to the development of NED in PCa [16]. EMT and NED may also be induced by androgen-targeted therapy [14] and are considered a resistance mechanism to treatments in PCa [17]. Due to re-activation of developmental programs with EMT induction and a stem cell phenotype, the resulting NE PCa (NEPC) is highly aggressive.

Interestingly, an inverse relationship between NED and EMT has also been described in some tumors, i.e., small cell lung carcinoma (SCLC), Merkel cell carcinoma, and gastroenteropancreatic (GEP)-NET [18]. In SCLC, an association was revealed between the loss of NED and EMT induction [19] as inferred from the observation that the low NED subtype had undergone EMT and had activated—amongst others—the TGF-β pathway. However, differential effects of TGF-β on both programs were observed in SCLC in that TGF-β seems to be required for promoting EMT but not NED. In the panNET cell lines, BON-1 (BON), QGP-1, and NT-3 previous work has shown that the ECAD and VIM expression profiles indicate a well-differentiated epithelial phenotype [20]. In BON cells, TGF-1 has been shown to control proliferation and NED through the SST/SSTR system [21,22]. Of note, disrupting either the TGF-β or SST signaling pathway resulted in NED-mesenchymal transition, which is characterized by the loss of NED markers, decreased ECAD, and elevated VIM expression. This inverse correlation of TGF-β signaling activity and EMT was surprising since TGF-β is known as one of the most potent inducers of EMT. In a transgenic mouse model of PCa, aberrant TGF-β signaling enhances EMT and NED, thereby driving tumor progression to CRPC [23].

Given the concept of a positive association between EMT and NED mainly arising from the PCa model, an important issue remains whether the reverse process, mesenchymal–epithelial transition (MET), affects NED. We have recently shown that PDAC-derived tumor cells can be forced to undergo MET in response to a transdifferentiation culture through exposure to a cocktail of three cytokines, IL-1β, IFN-γ, and TNF-α (TDC-IIT) [6]. Moreover, bone morphogenetic protein-7 (BMP-7), a member of the TGF-β superfamily of growth and differentiation factors, has been reported to be able to induce MET in adult renal fibroblasts of the injured kidney [24], hepatic stellate cells [25], and melanoma cells [26], generating functional epithelial cells [27]. Interestingly, BMP-7 has been shown to induce MET through downregulation of the EMT-related transcription factor SNAIL [28], SNAIL-induced α-smooth muscle actin, and concomitant upregulation of ECAD. BMP-7 also acts as an inhibitor of fibrotic progression in many organs through activation of the SMAD1/5 arm of TGF-β signaling and inhibition of TGF-β–mediated EMT [29,30] via suppression of canonical TGF-β/SMAD2/3 signaling [31].

Given the above-mentioned findings, particularly the inverse relationship between EMT and NED in panNET, we decided to study the effects of EMT and MET inducers on EMP and NED in pancreatic cells (normal duct cells and PDAC-derived tumor cells of the epithelial and quasi-mesenchymal subtype) and in cell lines of SCCOHT. Specifically, we asked whether forced conversion from epithelial to mesenchymal (via treatment with TGF-β1) or vice versa (via stimulation with TDC-IIT or BMP-7) will affect NED markers in the same way as EMT markers.

## 2. Material and Methods

### 2.1. Cells

PANC-1, MIA PaCa-2, and BxPC-3 human PDAC cells were obtained from the ATCC (Manassas, VA, USA) and propagated in RPMI 1640 (Sigma-Aldrich, St. Louis, MO, USA) containing fetal bovine serum (FBS, 10%), penicillin–streptomycin–glutamine (1%, Thermo Fisher Scientific, Darmstadt, Germany), and sodium pyruvate (1%, Merck Millipore/Sigma Aldrich, Taufkirchen, Germany). HPDE6c7 (HPDE) were purchased from AddexBio (#T0018001). BON cells were established from a functional human pancreatic carcinoid tumor and were originally provided by C.M. Townsend (University of Texas, Galveston, TX, USA). The NT-3 cell line has been established and characterized by our group in 2018 [32]. Maintenance of BON and NT-3 cells has been described in detail previously [20,22]. The generation of individual PANC-1 cell clones has been described previously [6]. Cells were counted with a Neubauer chamber. For determination of basal mRNA levels, several independent RNA isolates (3–6) were retrieved from continuous cultures of cells and subjected to qPCR analysis.

Established and characterized cellular models of SCCOHT are represented by the two cell lines BIN-67 (kindly provided by Dr. Barbara Vanderhyden, University of Ottawa, Canada) and SCCOHT-1. BIN-67 were cultured with DMEM/F12:DMEM medium (1:1, *v*/*v*) (Sigma Aldrich) supplemented with 20% FBS (PAN-Biotech GmbH, 94501 Aidenbach, Germany), L-glutamine (2 mM), penicillin (100 U/mL), and streptomycin (100 µg/mL) (all from Capricorn Scientific GmbH, Ebsdorfergrund, Germany) [11].

The SCCOHT-1 cells were isolated, processed, and cultured as a spontaneously growing primary culture derived from a tumor biopsy of a 31-year-old patient with recurrent SCCOHT [9]. Studies with these cells have been approved by the Ethics Committee of Hannover Medical School, Project #3916, on 15 June 2005, and prior informed written consent was obtained from the patient for the use of this material. SCCOHT-1 cells were maintained in 1640 supplemented with 10% FBS (PAN-Biotech GmbH), 100 U/mL of L-glutamine, 100 U/mL of penicillin, and 100 µg/mL of streptomycin (all from Capricorn Scientific GmbH). Cells were kept at 37 °C in a humidified atmosphere of 5% CO_2_, and the culture medium was changed every 3 to 4 d.

The cell lines were tested for mycoplasma by the luminometric MycoAlert Plus mycoplasma detection kit (Lonza Inc., Rockland, ME, USA) according to the manufacturer’s recommendations. Authentication of all cell lines used in this study was performed by short tandem repeat (STR) fragment analysis using the GenomeLab human STR primer set (Beckman Coulter Inc., Fullerton, CA, USA). STR fragments for the ovarian cancer cells were confirmed as described previously [33] (Appendix A).

In some experiments, cells were treated with varying concentrations of either recombinant human TGF-β1 (#300-023, ReliaTech, Wolfenbüttel, Germany) (0–10 ng/mL) or human BMP-7 (#120-03P, Preprotech, Hamburg, Germany) (0–200 ng/mL). The ALK4/5/7 inhibitor SB431542 and the BMP type I receptor inhibitor LDN193189 were purchased from Merck (Darmstadt, Germany).

### 2.2. QPCR Analysis

Total RNA was extracted and purified from cells using affinity chromatography on columns (innuPREP RNA Mini Kit 2.0, IST Innuscreen GmbH, Berlin, Germany) according to the manufacturer’s instructions. An amount of 2.5 μg of RNA per sample was subjected to reverse transcription in a total volume of 20 μL at 37 °C for 1 h using M-MLV reverse transcriptase (200 U) and random hexamers (2.5 μM) (Thermo Fisher Scientific). The relative expression of the genes of interest was quantified by quantitative real-time PCR on an I-Cycler (BioRad, Munich, Germany) using Maxima SYBR Green Mastermix (Thermo Fisher Scientific). Following the generation of C_t_ values for the target genes, these were normalized with those for either TATA-box-binding protein (TBP) or glyceraldehyde-3-phosphate dehydrogenase (GAPDH). The sequences of PCR amplification primers are given in Appendix A.

### 2.3. Cell Lysis and Immunoblotting

Subconfluent cultures of cells were lysed with 1× PhosphoSafe lysis buffer (Merck Millipore), and after sonication and clearing, total protein concentration of the supernatants was determined with the BioRad DC Protein Assay (BioRad). Samples were fractionated on mini-PROTEAN TGX any-kD precast gels (BioRad) and blotted to 0.45 μm PVDF membranes. Following blocking of the membranes with nonfat dry milk or BSA, their incubation with primary antibodies proceeded overnight at 4 °C. HRP-linked secondary antibodies and Amersham ECL Prime Detection Reagent (GE Healthcare, Munich, Germany) were used for chemiluminescent detection of proteins on a BioRad ChemiDoc XRS imaging system. We employed the following primary antibodies: anti-HSP90 (F-8), #sc-13119, and anti-vimentin, sc-6260 (Santa Cruz Biotechnology, Heidelberg, Germany); anti-RAC1b, #09-271 (Merck Millipore); anti-E-cadherin, #3195, anti-GAPDH (14C10), #2118, and anti-Snail, #3895 (Cell Signaling Technology, CST, Frankfurt am Main, Germany); anti-Synaptophysin and anti-human Chromogranin A, clone DAK-A3 (both from Dako, Glostrup, Denmark); anti-Somatostatin Receptor-2 antibody [UMB1]-C-terminal, #ab134152 (Abcam, Cambridge, UK); and anti-Phospho-Smad2 (Ser465/467) (CST) and anti-human Phospho-Smad3 (R&D Systems, Wiesbaden, Germany). HRP-linked anti-rabbit, #7074, and anti-mouse, #7076, secondary antibodies were from CST.

### 2.4. Immunohistochemistry (IHC)

Immunohistochemical staining was performed for two cases of PDAC (#88734: CA/PDAC, pancreatic head, pT2 pN2(4/15) L1 V0 G2, male, age 55 y, and #16680: poorly differentiated PDAC (4.7 cm) with infiltration of peri-pancreatic fat tissue and a segment of co-resected small intestine, pT3 pN0(0/13) L0 V1 Pn1 pR1 G3, male, age 65 y) as well as one metastasis (#16647: liver metastasis of a primary adenocarcinoma of the pancreas, pTx pNX pMx, female, age 65 y). FFPE tissue sections were stained for CHGA, SSTR2, and SYP according to a previously published protocol [34]. Briefly, the FFPE slides were deparaffinized by sequential immersion in xylene followed by rehydration through a graded ethanol series, antigen retrieval with citric acid-based buffer, and blocking for endogenous peroxidase. Anti-CHGA (1:100), anti-SYP (1:50), anti-SSTR2 (1:100), and mouse mabs to human CK7 (Dako, clone OV-TL #12/30, 1:200) and human vimentin (Abcam, V1-RE/1, #ab3974, 1:200) were applied to the tissue sections, and slides were incubated overnight at 4 °C. After incubation with secondary antibodies, visualization was achieved with the DAB Detection Kit (Biomol, Hamburg, Germany). Negative controls were generated by incubating adjacent sections with the respective unconjugated IgG.

### 2.5. Real-Time Cell Migration Assays

The impedance-based RTCA assay (xCELLigence^®^ technology, Agilent Technologies, Santa Clara, CA, USA, supplied by OLS, Bremen, Germany) was carried out exactly as described in detail earlier [6,22,35,36]. Briefly, CIM plates-16 coated on the underside with a 1:1 mixture of collagens I and IV to facilitate adherence of cells were loaded with 80,000 cells per well and either TGF-β1 (10 ng/mL), BMP-7 (200 ng/mL), or both. Changes in impedance were recorded every 15 min, and data were analyzed with the RTCA software (version 2.0, Agilent Technologies).

### 2.6. Statistical Analysis

Statistical significance was calculated with the unpaired Student’s *t* test. Results were deemed significant at *p* < 0.05.

## 3. Results

### 3.1. PANC-1 and MIA PaCa-2 Cells, but Not Normal Pancreatic Duct Epithelial Cells or Epithelial-Subtype Pancreatic Tumor Cells, Constitutively Express Various NED Markers

As mentioned in the Introduction, the quasi-mesenchymal cell lines PANC-1 and MIA PaCa-2 have been shown earlier to express various NED markers [3,4,5]. Here, we confirmed endogenous expression of the SYP, SSTR2, and SSTR5 genes in both cell lines by qPCR analysis, which was clearly higher in PANC-1 cells (Figure 1). This was most apparent for another prominent NED marker, CHGA, the mRNA of which was not detectable in MIA PaCa-2 cells. As positive controls, we employed the panNET cell lines, BON-1 and NT-3 [22] (Figure 1).

If NED segregates with a transformed/tumor cell phenotype, then HPDE, a normal (immortalized) cell line established from pancreatic ductal epithelial cells, should be devoid of expression of NED markers. Indeed, in qPCR analysis, HPDE cells were either negative or only weakly positive for all NED markers tested except SSTR5 (Figure 1), and this was confirmed for CHGA, SYP, and SSTR2 by immunoblot analysis. The PDAC-derived moderately differentiated cell line BxPC-3 was shown earlier to be strongly positive for ECAD but negative for VIM [6]. This suggests that these cells, despite their transformed state, have retained an epithelial phenotype, hence representing the classical/epithelial histomorphological subtype of PDAC [6]. Based on the postulated assumption that NED is associated with a mesenchymal phenotype, we reasoned that BxPC-3 cells should exhibit lower expression of SYP and CHGA. When compared to PANC-1, this was indeed the case (CHGA: 8-fold lower), SYP (400-fold lower) (Figure 1), and NSE (10-fold lower). However, only SSTR5 was expressed by BxPC-3 at the same level as PANC-1 (Figure 1). In contrast to CHGA and SYP, SSTR2 was also detectable by immunoblot analysis in BxPC-3 (Appendix A). For protein expression of SYP and SSTR2 in PANC-1 cells, see Figure 2, Figure 3 and Figure 4. We conclude that poorly differentiated PDAC cell lines, in particular PANC-1, but not their presumed non-transformed precursor cells, express moderate amounts of SYP and CHGA. Moderately differentiated PDAC cells—as exemplified by BxPC-3—exhibit lower mRNA levels of SYP and CHGA (but not SSTR2 and 5) when compared to PANC-1 cells.

### 3.2. NED Marker Expression Varies in Single-Cell-Derived Clones of PANC-1 Cells with Different EMT Phenotypes

We have shown recently that the PANC-1 cell line displays EMP; parental cultures consist of clonal subpopulations with different EMT phenotypes as evidenced by different ratios of ECAD:VIM expression in single-cell-derived clones [6]. This unique feature allowed us to study the association between EMT and NED within a genetically identical background. Here, we analyzed seven clones, exhibiting either a more epithelial (E) or mesenchymal (M) phenotype, initially by qPCR analysis for expression of CHGA, SYP, NCAM, NSE, GLUT2, SSTR2, and SSTR5. Results indicate that clones with a low ECAD/VIM ratio (M-type: P1C3, P3D10, P4B9, P2E8) when compared to clones with a high ECAD/VIM ratio (E-type: P4B11, P3D2, P1G7) present with higher levels (mean ± SD) of CHGA (5.21 ± 2.1 vs. 2.26 ± 1.12), SYP (1.18 ± 0.44 vs. 0.811 ± 0.16), NSE (1.28 ± 0.35 vs. 1.16 ± 0.14), NCAM (2.04 ± 0.92 vs. 0.86 ± 0.16), SSTR2 (1.49 ± 0.42 vs. 0.92 ± 0.10), and SSTR5 (2.90 ± 1.34 vs. 0.98 ± 0.40) (Figure 2A). Yet another marker, GLUT2, was not different between M and E-type cells (0.86 ± 0.42 vs. 0.90 ± 0.19) (Figure 2A). We then performed immunoblot analysis of SYP and SSTR2. Of note, the abundance of both the SYP (Figure 2B) and SSTR2 proteins mirrored their mRNA levels (high in M-type clones P1C3, P3D10, and P4B9, low in E-type clones P1G7, P3D2, and P4B11). We conclude that NED is more pronounced in clonal subcultures with an M-phenotype when compared to those with an E-phenotype.

### 3.3. IHC of NED Markers in PDAC Tissue Specimens from Surgically Resected Patients

Next, we sought to identify the sites of CHGA, SSTR2, and SYP protein expression in various PDAC specimens from surgically resected patients using IHC. For this purpose, we selected primary pancreatic tumors of different size/extent, namely T2 (invasion of the muscularis propria) and T3 (invasion into the subserosa), and differentiation grade, G2 (moderately differentiated) and G3 (poorly differentiated), as well as a liver metastasis. In all specimens, CHGA, SSTR2, and SYP exhibited strong staining in the (ductal) epithelial tumor cells (Figure 3). In addition, weaker signals for CHGA and SSTR2 but not SYP proteins were also noted in cells of the surrounding stroma (Figure 3).

SNAIL, SLUG, and ECAD have been demonstrated previously by IHC to be expressed by the ductal tumor cells in primary human PDAC tissue specimens and orthotopic tumors derived from transplantation of MIA PaCa-2 and PANC-1 cells into nude mice [37]. We therefore stained adjacent sections from the same PDAC specimens for two other EMT markers, CK7 and VIM. CK7 was chosen because it is overexpressed in most cancers, including PDAC, and is associated with increased proliferation, migration, metastasis, and TGF-β-induced EMT [38]. CK7 was exclusively expressed in the ductal tumor cells, while VIM was present in most tumor cells lining the ducts and in stromal fibroblasts (Figure 3). From these data and those from an earlier study [37], we conclude that NED and EMT markers are co-expressed by the neoplastic ductal epithelial cells from PDAC.

### 3.4. Treatment of PANC-1 Cells with TGF-β1 or BMP-7 Alters EMT- and NED-Associated Gene Expression

Prompted by the crucial role of TGF-β in pancreatic NE tumors [21] and the known existence of a PDAC subtype exhibiting NED, we sought to study the response of PANC-1 cells to TGF-β1 and BMP-7. The former growth factor is a strong inducer of EMT, while the latter is an inhibitor of EMT, a promoter of MET [22,23,24,25], and a presumed suppressor of canonical TGF-β/SMAD2/3 signaling [29]. In PANC-1 cells, treatment with TGF-β1 or, surprisingly, BMP-7 downregulated ECAD and another epithelial marker, RAC1b (a tumor-associated splice isoform of human *RAC1*) [39]. At the same time, both growth factors upregulated the mesenchymal markers SNAIL1 (SNAIL), SNAIL2/SLUG, and VIM, while another mesenchymal marker, RAC1, remained unaltered (Figure 4A). The effect of TGF-β1 was more potent than that of BMP-7 for RAC1b, SNAIL, and SLUG, while no significant differences between both growth factors were seen for RAC1, ECAD, and VIM (Figure 4A). The simultaneous co-treatment with TGF-β1 and BMP-7 acted in an additive manner to suppress ECAD and to enhance SNAIL, SLUG, and VIM (Figure 4A).

The regulatory effects of TGF-β1 and BMP-7 on EMT markers at the mRNA level were reproduced for the ECAD, RAC1b, SNAIL, and VIM proteins as shown by immunoblotting (Figure 4B). This suggested that both TGF-β1 and, unexpectedly, also BMP-7 induce EMT in PANC-1 cells and that BMP-7 can synergize with TGF-β1 in EMT induction.

Next, we monitored in TGF-β1 or BMP-7-treated parental cultures of PANC-1 cells the expression of NED-associated genes. Immunoblotting revealed that SYP protein levels were induced by both TGF-β1 and BMP-7 (Figure 4B). Moreover, combined treatment of PANC-1 cells with TGF-β1 and BMP-7 had a synergistic effect on SYP protein abundance (Figure 4B). However, unlike TGF-β1, BMP-7 failed to induce SYP mRNA (Figure 4C), suggesting that regulation of SYP by BMP-7 involves alterations in protein stability/half-life rather than de novo transcription. In contrast to SYP, SSTR2 mRNA (Figure 4C) was induced by both TGF-β1 and BMP-7, with the induction by TGF-β1 being greater than that by BMP-7 (Figure 4C, *p* = 0.023). Intriguingly, the combined treatment of PANC-1 cells with TGF-β1 and BMP-7 had a synergistic effect on SSTR2 but not on SYP mRNA levels (Figure 4C). In contrast to SSTR2, SSTR5 was strongly downregulated by TGF-β1 and TGF-β1+BMP-7 but, surprisingly, was upregulated by BMP-7 alone (Figure 4C). In addition, TGF-β1 treatment induced *CHGA*, *NCAM*, and *NSE* (Figure 4C). In contrast, BMP-7 suppressed *CHGA* (Figure 4C) much like TGF-β1 induced *NSE* (Figure 4C). For a selected panel of genes, which have been shown above to be either upregulated (SSTR2) or downregulated (RAC1b, SSTR5) by TGF-β1 in parental cultures of PANC-1 cells, we also evaluated whether the TGF-β1 effects vary among individual clones. Analysis of five clones indeed showed that these responded differently with respect to induction of SSTR2 and suppression of RAC1b and SSTR5, but not RAC1 (Figure 4D).

We explored the effects of varying concentrations of TGF-β1 or BMP-7 on *SNAI1* mRNA in PANC-1 cells to determine whether these are dose-dependent (Appendix A). Moreover, the ability of the TGF-β/ALK5 signaling inhibitor SB431542 [40] and the BMP-7/ALK2 inhibitor LDN193189 [36] to prevent upregulation of *SNAI1* by TGF-β1 or BMP-7, respectively (Appendix A), strongly suggests that these inducive effects are indeed driven by TGF-β and BMP receptor activation.

### 3.5. Treatment of BxPC-3 and HPDE Cells with TGF-β1 or BMP-7 Alters EMT- and NED-Associated Gene Expression, While MIA PaCa-2 Cells Only Responded to BMP-7

MIA PaCa-2 cells have been reported to be insensitive to TGF-β stimulation due to a lack of expression of the TGF-β type II receptor [41]. Accordingly, MIA PaCa-2 cells failed to respond with any changes in EMT or NED marker expression to treatment with TGF-β1 alone (Figure 5A), but whether these cells can respond to BMP-7 was unknown. As in PANC-1 cells, treatment with BMP-7 (200 ng/mL) induced the mRNA expression of SNAIL (x1.31), SLUG (x2.91), VIM (x1.41), and SSTR5 (x2.72) (Figure 5A), while no significant changes were seen for RAC1, RAC1b, SYP, and SSTR2. Both ECAD and CHGA were undetectable in MIA PaCa-2 even by qPCR.

Expression of SNAIL and VIM in MIA PaCa-2 cells was also analyzed by immunoblotting. SNAIL, but not VIM, was induced by BMP-7 alone; however, when combined with TGF-β1, BMP-7 was able to also increase VIM levels (Figure 5B). As expected from the high C_t_ values in qPCR analysis of MIA PaCa-2 cells, SYP and SSTR2 were below detection limits in immunoblots.

Next, we studied the response of BxPC-3 and HPDE cells to challenge with TGF-β1 and BMP-7. Both BxPC-3 (Appendix A) and HPDE (Appendix A) responded only moderately or not at all to these growth factors with respect to EMT (ECAD, SLUG, SNAIL, VIM) and NED (SSTR2, SSTR5) markers and rarely exhibited additive or synergistic effects on simultaneous treatment with both growth factors.

The data presented in Figure 4 and Figure 5, Appendix A revealed that in quasi-mesenchymal PANC-1 and MIA PaCa-2, but not BxPC-3 and HPDE cells, TGF-β1 repressed epithelial genes and strongly induced mesenchymal and NED genes, except for SSTR5. In contrast, BMP-7 differentially impacted gene expression; it synergized with TGF-β1 in the induction of *VIM*, *SNAI1*, *SSTR2*, and *ENO2* (encoding NSE) but antagonized it in the regulation of *SSTR5* and *CHGA* (if expressed).

### 3.6. Analysis of Smad Signaling by TGF-β1 or BMP-7 in PANC-1 Clones

In Figure 4 and Figure 5 we showed that parental PANC-1 cells and five PANC-1 subclones responded to TGF-β1 stimulation with regulation of various EMT and NED markers. TGF-β signals through the type I receptor ALK5 and the receptor-regulated Smad proteins (R-Smads) SMAD2 and SMAD3, while BMP-7 signals through ALK2 and the R-Smads SMAD1 and SMAD5 [42]. The R-Smads are activated by the activated type I receptor kinases through phosphorylation at serines in a conserved C-terminal SSXS motif [42]. Following treatment of two PANC-1 clones (P1C3, M-type, and P3D2, E-type) for 1 h with either TGF-β1 or BMP-7, or a combination of both growth factors, phospho-immunoblotting was performed to assess the levels of activated SMAD2 and SMAD3. Interestingly, both clones responded to TGF-β1 and TGF-β1+BMP-7 with strong C-terminal phosphorylation of SMAD2 and SMAD3 (Figure 6). Of note, BMP-7 alone also induced a small increase in the abundance of p-SMAD3 (in both clones) and p-SMAD2 (only in P3D2, Figure 6), which indicates that BMP-7 can trigger TGF-β-like signaling, and through this ability may promote EMT and NED. Moreover, in P1C3, but not P3D2 cells, the combination of TGF-β1 and BMP-7 dramatically and synergistically enhanced the signal strength for p-SMAD3 but not p-SMAD2 over that of TGF-β1 alone (Figure 6). This is an intriguing observation given the additive or synergistic effects of the TGF-β1+BMP-7 co-treatment on the expression of some EMT- and NED-associated genes (see Figure 4).

### 3.7. The Expression of NED Markers Is Altered During MET and Positively Correlates with That of Mesenchymal Markers

Our previous studies have shown that poorly differentiated quasi-mesenchymal PANC-1 and MIA PaCa-2 cells can be induced to undergo MET by a 72 h treatment with IL1-β, IFN-γ, and TNF-(IIT) [6]. This transdifferentiation culture (TDC) of both cell lines with IIT (TDC-IIT) was associated with reversal of EMT, as evidenced by upregulation of epithelial markers ECAD, CLDN4, CK19, GRHL2, and OVOL2 [6]. Here, we noted, in addition, an induction of RAC1b mRNA in PANC-1 (but repression in MIA PaCa-2, Figure 7A) and a decline in RAC1, VIM, and SNAIL mRNAs in both cell lines (Figure 7A).

Given the reciprocal pattern of regulation of epithelial and mesenchymal markers in MET compared to EMT, it was of interest to analyze if exposing parental cultures of PANC-1 or MIA PaCa-2 cells to TDC-IIT also alters NED gene expression. Based on the positive association of NED and mesenchymal genes (see above), we reasoned that the expression of NED genes should decrease in response to the MET-inducing conditions. Intriguingly, using qPCR analysis, we observed in PANC-1 cells strong downregulation of CHGA, SYP, NCAM, NSE, and SSTR5 but not SSTR2 (Figure 7B, upper graphs). However, immunoblots indicated a decline in protein levels of SSTR2 (Appendix A), which was not unexpected given the upregulation by the EMT inducers TGF-β1 and BMP-7 (see Figure 4). In MIA PaCa-2 cells, we noted downregulation of SYP, NSE, and SSTR5 and upregulation of NCAM, SSTR2 (Figure 7B, lower graphs), and GLUT2. This clearly shows that the major EMT and NED markers are co-regulated and that MET is associated with a loss of both the mesenchymal and NED phenotypes.

### 3.8. Cell Lines Derived from SCCOHT Differentially Respond to TGF-β1 or BMP-7 Treatment with Upregulation of EMT and NED Markers

Two different cell lines, BIN-67 and SCCOHT-1, representing cellular models for SCCOHT [9,10,11], were employed to test possible effects of TGF-β1 and BMP-7 on EMT- and NED-associated genes. An initial qPCR-based screening indicated that BIN-67 and SCCOHT-1 cells only express ECAD, SNAIL, VIM, CHGA, SYP, and SSTR2 to a significant extent. Other genes (SLUG, NCAM, GLUT2, and SSTR5) were either negative or only weakly positive, while NSE was expressed only in SCCOHT-1 but not in BIN-67 cells.

Both cell lines were then treated for 24 h with either TGF-β1, BMP-7, or a combination of both and subjected to qPCR analysis of the above EMT and NED markers. Results show that BIN-67 cells were refractory to stimulation with either growth factor for the above-mentioned genes, except ECAD, which was induced by BMP-7 by a factor of 1.6.

In SCCOHT-1 cells, both growth factors were unable to alter the expression of ECAD, SYP, or NSE. However, *SNAI1* failed to respond to TGF-β1 treatment but was induced 1.82-fold (*p* = 0.02) by BMP-7 and 2.33-fold (*p* = 0.002) by TGF-β1+BMP-7 (Figure 8). The enhancing effect of the combination of TGF-β1+BMP-7 over BMP-7 alone was statistically significant (*p* = 0.045) (Figure 8). Likewise, although VIM was upregulated 1.2-fold (*p* = 0.065) by BMP-7, only the combined treatment generated a statistically significant induction (1.3-fold, *p* = 0.004, Figure 8). The only NED markers that were responsive to BMP-7 were CHGA (1.35-fold, *p* = 0.017) and 1.59-fold (*p* = 0.031) by TGF-β1+BMP-7, and SSTR2 (1.78-fold, *p* = 0.007) by BMP-7, and 2.13-fold (*p* = 0.021) by TGF-β1+BMP-7 (Figure 8). We conclude that both cell lines were unresponsive to TGF-β1 (even in response to a high concentration such as 10 ng/mL), while SCCOHT-1, but not BIN-67, cells have retained sensitivity to BMP-7. Intriguingly, although resistant to single treatment, TGF-β1, when used in combination with BMP-7, was able to enhance the BMP-7 effect on SNAIL (Figure 8). For VIM, CHGA, and SSTR2 we observed the same trend, but the differences between BMP-7 and TGF-β1+BMP-7 missed statistical significance.

### 3.9. Treatment of PANC-1 and SCCOHT-1 Cells with TGF-β1 and/or BMP-7 Alters Migratory and Proliferative Activities

As shown in Figure 4, known promoters of invasion (SNAIL, SLUG, and VIM) were upregulated and known repressors of invasion (ECAD and RAC1b) downregulated by TGF-β1, BMP-7, or both. Moreover, phospho-immunoblotting indicated activation of SMADs 2 and 3 by both TGF-β1 and BMP-7 (see Figure 6), which should result in alterations in cellular function. We, therefore, measured two cancer-relevant responses, cell migration/invasion and proliferation. Using real-time cell analysis (RTCA) of cell migration of PANC-1 cells, we observed that both TGF-β1 and BMP-7 independently stimulated migratory activities and that when applied together, they displayed an additive/synergistic effect (Figure 9A).

Finally, we determined the proliferative activities of SCCOHT-1 and BIN-67 cells in response to a challenge with TGF-β1 and/or BMP-7. Cell counting assays indicated a significant reduction in cell counts of BIN-67 after a 24 h treatment with BMP-7, while TGF-β1 had no effect. In SCCOHT-1 cells, a decrease in cell numbers was seen only in cells treated with TGF-β1 and BMP-7 simultaneously (Figure 9B). These data confirm the refractoriness of both cell lines to TGF-β and, in addition, reveal a growth-inhibitory function of BMP-7.

## 4. Discussion

The relationship between EMT and NED, and the role of TGF-β signaling in controlling these differentiation programs, appears to vary among different tumor entities and even between cancers affecting the same organ, such as the pancreas. While these associations have been studied in detail in a panNET cell line [21], only little information was available for PDAC in this respect. For other tumor entities, such as SCCOHT, no such data, whatsoever, were available. We thus initially focused on the pancreatic model, comparing poorly differentiated tumor cells of the quasi-mesenchymal subtype (PANC-1, MIA PaCa-2) with moderately differentiated ones of the epithelial subtype (BxPC-3) and with the presumed progenitor cells, non-transformed pancreatic ductal epithelial cells (HPDE). Based on our assumption that NED is associated with EMT and a mesenchymal phenotype, we hypothesized that HPDE and BxPC-3 cells should not exhibit a NED phenotype. Indeed, HPDE expressed much lower levels of NED-associated markers than PANC-1 and MIA PaCa-2. The same was true for BxPC-3 with respect to CHGA, SYP, and NSE, although, unlike HPDE, not for SSTR2 and SSTR5. Hence, we have shown in PANC-1 and MIA PaCa-2 cells endogenous expression of NED markers, some of which had previously been detected by flow cytometry [3].

PANC-1 cells exhibit EMP, meaning that parental cultures of this cell line consist of a mixture of subclones, each displaying a different EMT phenotype despite being genetically identical. A previous histomorphological subtyping with a panel of epithelial and mesenchymal markers has shown that these clones can be grossly classified as epithelial (E-type), mesenchymal (M-type), or mixed [6]. We, therefore, considered it appropriate to test whether NED markers are enriched in either the E- or the M-type clones. Monitoring seven (three E-type and four M-type) single cell-derived clones for expression of SYP, CHGA, NCAM, NSE, GLUT2, SSTR2, and SSTR5 showed that M-type clones (P1C3, P3D10, P4B9, and P2E8) present with higher levels of these NED markers, except for GLUT2, than E-type clones (P4B11, P3D2, and P1G7). This led us to conclude that NED is preferentially associated with a mesenchymal phenotype.

Next, we sought to know if TGF-β1, a powerful promoter of EMT, and BMP-7, another member of the TGF-β superfamily of growth and differentiation factors and promoter of MET, impact EMT- and NED-associated gene expression in pancreatic tumor cells. While PANC-1 cells are highly sensitive to this growth factor, MIA PaCa-2 cells are refractory due to a defective type II receptor [41]. These cells could thus only be employed to study the effects of BMP-7. Treatment of parental PANC-1 cells with TGF-β1 downregulated ECAD and RAC1b and upregulated VIM and SNAIL. However, both PANC-1 and MIA PaCa-2 cells responded to treatment with BMP-7 with induction of SNAIL and VIM, while RAC1b was only suppressed in PANC-1 and RAC1 remained unaltered in both cell lines. The effect of TGF-β1 on VIM and SNAIL was more potent than that of BMP-7. Co-treatment with both growth factors acted in either an additive or synergistic manner to suppress ECAD (only in PANC-1 since MIA PaCa-2 are ECAD-null) and RAC1b and to enhance SNAIL and VIM expression at both the RNA and protein levels. We thus concluded that TGF-β1 and BMP-7 in PANC1 and BMP-7 in MIA PaCa-2 cells can induce EMT and that BMP-7 can synergize with TGF-β1 in EMT induction. The observation that BMP-7 promoted EMT was surprising since this growth factor has been identified in other cellular models as either an inhibitor of EMT or even a promoter of MET [24,25,26,27,28].

Both TGF-β1 and BMP-7 were capable of inducing SYP protein in PANC-1; however, only TGF-β1 appears to accomplish this by a transcriptional mechanism. Intriguingly, a very strong inductive effect of TGF-β1 or BMP-7 was observed on SSTR2 in PANC-1 cells that could be further enhanced synergistically by combined treatment. In MIA PaCa-2 cells, however, SSTR2 mRNA levels remained unchanged in response to TGF-β1 or BMP-7, while the related SSTR5 was strongly downregulated by TGF-β1 but upregulated by BMP-7 in both cell lines. In addition, TGF-β1 treatment of PANC-1 induced CHGA, NCAM, and NSE, while BMP-7 treatment only upregulated NSE but downregulated CHGA. These results clearly indicate that at least in control of SSTR5 and CHGA, BMP-7 can also act in an antagonistic fashion to TGF-β1.

SSTR2 and -5 do not only represent established markers of NED but also possess tumor-relevant functions. SSTR2 is an inhibitory G protein-coupled receptor, the expression of which is lost in most human pancreatic cancers [43]. Of note, murine Sstr2 has been identified as a transcriptional target of TGF-β [35], which suggested the possibility that loss of SMAD4 accounts for the loss of SSTR2 expression in human PDAC. This event may contribute to a growth advantage of tumor cells [43] and is consistent with findings that SSTR2 exhibits anti-tumor properties. Here, we have confirmed—for the first time—in human PDAC-derived tumor cells a strong positive regulation of SSTR2 by TGF-β1. Also, for the first time, *SSTR5* was identified here as a negative transcriptional target gene of TGF-β1 as evidenced by the dramatic downregulation of its mRNA. This mode of regulation suggests the possibility that SSTR5 normally antagonizes TGF-β-dependent EMT or even other cellular responses to TGF-β, such as growth arrest. Hence, while *SSTR2* qualifies as a gene involved in growth arrest in accordance with the proposed anti-tumor function, the reverse may be true for *SSTR5*. Moreover, while SSTR2 appears to be involved in mesenchymal conversion, SSTR5 may have a role in promoting MET or an epithelial phenotype. An interesting finding in this context came from a study with a highly invasive paclitaxel-resistant OC cell line. This cell line expresses CD105/endoglin, a stem cell marker and TGF-β co-receptor that may promote EMT and metastasis of OC by inhibiting expression of ECAD. Of note, after CD105 knockdown, the expression of both SSTR5 and ECAD (amongst others) was markedly upregulated [44]. Conversely, coactivation of SSTR2 in PDAC cells led to increased expression of mesenchymal markers and decreased expression of an epithelial marker [45]. Moreover, the expression of SSTR2 (along with those of SNAIL, SLUG, and VIM) was associated with invasive non-functioning NETs of the pituitary [46].

Treatment of PANC-1 or MIA PaCa-2 cells with BMP-7 downregulated ECAD and upregulated VIM and SNAIL. This pro-EMT effect was quite surprising, as BMP-7 has been identified previously as a MET-inducing (and thus anti-EMT) factor in a range of different cell types, such as alveolar type II cells, adult renal epithelial tubular cells and fibroblasts, hepatic stellate cells, and melanoma cells [24,25,26,27,28]. Moreover, BMP-7 stimulation of PANC-1 or MIA PaCa-2 cells also altered the expression of some NED genes, i.e., *SSTR2*, in the same way as the EMT-associated genes. The SSTR2 mRNA was induced by BMP-7, although the extent of induction was not as great as that with TGF-β1. In contrast to the suppressive effect of TGF-β1, BMP-7 upregulated SSTR5 mRNA, which would be consistent with a pro-epithelial effect based on the above proposed function for SSTR5. However, the observation that BMP-7 induced only a few NED markers, while others were either inhibited or remained unaffected, questions its role as a general promoter of NED. Surprisingly, upon combined stimulation of PANC-1 cells with TGF-β1 and BMP-7, additive effects on induction of *SSTR2* were noted. Together, this clearly suggests an association of EMT and NED through TGF-β signaling, while BMP-7 only partially shares this ability in common with TGF-β.

Prompted by the newly discovered positive association of EMT and TGF-β signaling with NED, we evaluated in another set of experiments the possibility that, conversely, MET in PANC-1 cells is associated with a loss of NED. In agreement with this hypothesis, we observed that during IIT-induced MET, most NED markers, except SSTR2, were downregulated along with VIM, RAC1, and SNAIL at the mRNA (Figure 5) and protein [6] levels, while concomitantly, the epithelial markers ECAD, CLDN4, GRHL2, OVOL2, CK19, and RAC1b were all upregulated [6]. This clearly shows a simultaneous loss of NED and mesenchymal markers during MET and the acquisition of an epithelial phenotype. Mechanistically, this may—at least in part—be mediated through inhibition of the SMAD2/3 arm of TGF-β signaling, since all markers that are responsive to TGF-β1 treatment were also affected by TDC-IIT but in an antagonistic fashion.

The unexpected observation that BMP-7 positively regulated all EMT and some NED markers prompted us to study the underlying mechanism. Since TGF-β transduces signals via Smads 2 and 3, we examined the activation state of these Smads in two clones of PANC-1, one M-type (P1C3) and one E-type (P3D2), by phospho-immunoblotting and, as expected, observed strong activation of both Smads by TGF-β1. Surprisingly, however, although BMP-7 was able to weakly induce p-SMAD3 only in P1C3 cells, in combination with TGF-β1 it was able to synergistically enhance the p-SMAD3 (but not the p-SMAD2) signal over that induced by TGF-β1 alone (Figure 6). If this phenotype-specific difference can be confirmed in additional clones, it may explain—at least in part—the pro-EMT (rather than the well-known pro-MET) function of BMP-7, including the strong stimulatory effect on basal and TGF-β1-dependent migratory activity of parental PANC-1 cells.

It should be mentioned that BMP-7 has previously been reported to induce EMT in PDAC-derived cells, including PANC-1 [47]. The authors of this study have also demonstrated that BMP-7 enhances the cells’ invasiveness through an increase in the expression and activity of MMP-2; however, crosstalk with TGF-β/SMAD2/3 signaling has not been analyzed in this study [47]. Induction of EMT by BMP-7 has also been described in other cancer cells, i.e., PC-3 PCa cells [48], B16 mouse and A2058 human melanoma cells [49], and in normal airway epithelial cells during restitution of an injured epithelium [50].

As a control for cells with a strong NED phenotype, we employed the panNET cell lines, BON and NT-3, which are both epithelial in nature [19]. This is in sharp contrast to PDAC cells with NED, which are poorly differentiated/quasi-mesenchymal. In this study, we have, therefore, revealed fundamental differences between two major types of pancreatic cancer, PDAC and panNET, with respect to the association of NED with EMT and TGF-β signaling. In panNET, NED is associated with a well-differentiated epithelial phenotype and functional TGF-β and SST signaling, and defective TGF-β/SST signaling causes loss of NED with mesenchymal conversion [21]. However, in our study the reverse situation is operating; NED occurs in poorly differentiated mesenchymal cancer cells, which matches our findings on the co-localization of CHGA, SSTR2, and SYP with CK7 and VIM in the neoplastic ductal cells in tissue sections of PDAC specimens, in particular a large (T3) and poorly differentiated (G3) tumor (Figure 3). In contrast to CK18/19, CK7 is known to be highly expressed in various cancers, including PDAC, and is associated with increased proliferation, migration, metastasis, and TGF-β-induced EMT [38].

NED can still be further enhanced by activation of TGF-β or BMP-7 signaling, provided the cells have retained responsiveness to these growth factors. Thus, the role of SST/SSTR in TGF-β signaling may differ between panNET (BON, NT-3) and PDAC, which is also supported by the antagonistic regulation of *SSTR5* by TGF-β1. We have thus identified a novel distinguishing feature between panNET and PDAC. Consequently, it will be highly intriguing to test these properties in panNEC, a pancreatic cancer entity that combines features of both panNET and PDAC. Intriguingly, we have shown recently that not only treatment with TGF-β but also the stimulation with SST or the SST analogs, octreotide and lanreotide, was able to regulate a set of NED genes and alter the NED state [22].

While the phenotypic association between EMT and NED seems to be well established in some cancers, this is not the case for the underlying molecular mechanism(s). Initial insights came from the PCa model [51] with the identification of microRNA-147b as an inducer of NED through targeting the ribosomal protein PRS15A [52]. MicroRNA patterns may be altered by the exchange of exosomes between cancer cells and cells of the tumor microenvironment [53]. More recently, activation of NFκB-STAT3 signaling by tumor protein D52, isoform 3 (TPD52) has been found to induce distinct NED features (as measured by CHGA and NSE) through EMT under androgen-depleted conditions [51]. Moreover, the authors were able to show that TPD52 also positively regulates EMT of PCa cells towards NED (as revealed by induction of N-cadherin, VIM, and ZEB1, another EMT-associated transcription factor) via activation of NFκB-STAT3. These changes were orchestrated by SNAIL, since silencing of *SNAI1* in TPD52-positive cells blocked the progression of NED [51]. SNAIL may thus promote tumor aggressiveness in PCa cells through multiple processes: induction of EMT to promote migration, while, in turn, induction of NED promotes tumor proliferation through a paracrine mechanism [54]. In addition, ZEB1 has been shown to promote NED in PCa [55]. Liu and colleagues investigated the molecular mechanisms by which androgen deprivation therapy (ADT) induces NED in advanced PCa and found transmembrane protein 1 (MCTP1) to be abundantly expressed in samples from patients with advanced PCa. Of note, after ADT, MCTP1 through SNAIL promoted EMT, NED, and cell migration of PC-3 and C4-2 PCa cells [56].

Therapeutic targeting of SNAIL or ZEB1 may thus prove beneficial in abrogating not only EMT but also NED [54]. Apart from NFκB-STAT3 [51], other signaling pathways are involved in acquiring NED characteristic features of PCa cells, such as AMPK/SIRT1-p38MAPK-IL6 [57] and ERK [58]. Of note, MEK-ERK signaling is also activated by TGF-β and is critically involved in driving TGF-β-dependent EMT, migration, invasion, and metastasis [59,60,61], and TGF-β1-induced downregulation of *CDH1* and upregulation of *SNAI1* [39]. We are currently carrying out ERK immunoblot analysis of PANC-1 subclones to reveal whether the extent of ERK1/2 activation corresponds with NED marker expression. Interestingly, treatment of MIA PaCa-2 and PANC-1 cells with grape seed proanthocyanidins (GSPs) resulted in decreased phosphorylation of ERK1/2, inactivation of NFκB, reversal of EMT, upregulation of ECAD, downregulation of NCAD and VIM, and reduced cell migration [62]. It is thus conceivable that GSP-induced inhibition of ERK activation also reduces NED in these cells.

The importance of EMT is also considered in other GI cancers such as coloNEC. Current efforts are therefore underway to reveal an association between EMT and NED in model cell lines of this disease, e.g., the coloNEC-derived cell lines, SS-2 and LCC-18. Here, we have studied a non-GI cancer, namely small cell hypercalcemic ovarian cancer, represented by the cell lines SCCOHT-1 [9] and BIN-67 [10,11]. SCCOHT-1 cells constitutively express NED markers, such as NCAM, in the original patient tumor and derived mouse xenograft tumors [9,12]. Moreover, both cell lines developed biallelic deleterious SMARC A4 gene mutations whereby phenotypic and genetic similarities were observed between SCCOHT and highly malignant childhood-onset atypical teratoid/rhabdoid tumors (AT/RTs) of the central nervous system [33,63]. This heterogeneity and plasticity of SCCOHT-1 indicates the potential for transdifferentiation involving EMT and maturation along a NED phenotype [13]. We found both cell lines to be refractory to stimulation with TGF-β1 for any of the above-mentioned EMT/NED genes, but they have partially retained sensitivity to BMP-7 as evidenced by induction of ECAD in BIN-67 and SNAIL, VIM, CHGA, and SSTR2 in SCCOHT-1 cells.

We also performed functional assays to assess the physiological meaning of the TGF-β1 and BMP-7 effects. Specifically, we monitored migratory activity in parental PANC-1 cells and proliferative activity in BIN-67 and SCCOHT-1 cells in response to single and combined treatment with TGF-β1 and BMP-7. Both TGF-β1 and BMP-7 independently stimulated migratory activity, and the combination provided an additive or synergistic effect. It is interesting to note that SYP was upregulated along with EMT markers known to promote invasive activity (SNAIL, SLUG, VIM). This raises the issue of whether SYP, too, operates as an invasion promoter, which is currently being tested in our laboratory. Circumstantial evidence in favor of this possibility comes from other cancer types, such as gastric cancer [64] and PCa [65]. With respect to proliferative activities of SCCOHT-1 and BIN-67 cells in response to TGF-β1 and BMP-7 challenges, BIN-67 and cells exhibited a reduction in proliferation in response to BMP-7, while for SCCOHT-1 a decrease in cell numbers was seen only in cells treated with TGF-β and BMP-7 simultaneously. The inability of TGF-β1 to alter proliferation is consistent with its inability to alter EMT and NED marker expression, while the reverse is true for BMP-7 (BIN-67) or the combination of TGF-β1 and BMP-7 (SCOOHT-1).

Since both PANC-1 and MIA PaCa-2 cells are highly invasive and metastatic [66], the NED phenotype, besides the mesenchymal subtype, may contribute to this aggressive behavior. In addition, the TGF-β1 effects on various NED markers may have therapeutic significance, as it was shown that PANC-1 and MIA PaCa-2 cell lines, when subjected to (fractionated) radiation, upregulate not only the expression of NED genes [5] but also induce the synthesis and secretion of TGF-β [67]. Both radiotherapy and chemotherapy induce TGF-β activity, possibly promoting metastatic progression, and high levels of TGF-β are associated with resistance to anticancer treatments [68]. Therefore, irradiation-induced secretion of TGF-β1 by tumor cells may account for changes in NED marker expression. Given that the resulting induction of EMT and NED may enhance tumor invasion and metastasis in PDAC, the concomitant application of TGF-β inhibitors between radiation cycles should be considered to prevent an unwanted increase in tumor aggressiveness [68].

## 5. Conclusions

In the light of findings indicating an inverse relationship between EMT and NED in panNET and non-pancreatic tumors, we sought to reveal how the EMT inducers TGF-β1 and BMP-7 would affect EMT and NED in PDAC and in SCCOHT, a rare ovarian tumor. Employing established PDAC cell lines of the quasi-mesenchymal (PANC-1, MIA PaCa-2) or epithelial (BxPC-3) subtype, normal pancreatic duct epithelial cells (HPDE), and clonal cultures of PANC-1 exhibiting either a more mesenchymal (M) or epithelial (E) phenotype, we observed that the NE phenotype is more pronounced in quasi-mesenchymal cells and M-type PANC-1 clones. Moreover, treatment of the above cells with TGF-β1 or BMP-7 induced co-expression of EMT and NED markers and in PANC-1 cells stimulated migratory activity in vitro. Intriguingly, combined treatment with both growth factors enhanced the expression of EMT- and NED-associated genes as well as migratory activity in an additive or even synergistic manner. Conversely, inducing MET in PANC-1 cells by means of an inflammatory cytokine cocktail was associated with downregulation of mesenchymal and NED markers and upregulation of epithelial markers. Biochemical analysis of PANC-1-derived M- and E-type clones showed that BMP-7 synergistically amplified TGF-β1-specific SMAD3 signaling only in M-type cells, possibly providing a mechanistic explanation for the synergistic effect of BMP-7 on TGF-β1-induced gene expression and invasion in (quasi-)mesenchymal cells. We conclude that in PDAC, mesenchymal and NE markers are co-regulated, suggesting that both EMT and NED contribute to the aggressive phenotype of this cancer type.

## Figures and Tables

**Figure 1 cells-13-02010-f001:**
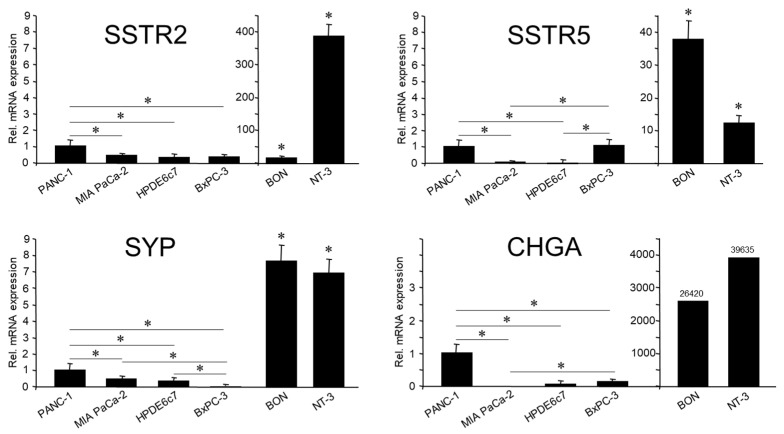
Relative expression of NED markers in PDAC-derived cell lines, PANC-1, MIA PaCa-2, and BxPC-3, and in the normal pancreatic duct epithelial cell line, HPDE. The panNET-derived cell lines, BON and NT-3, were employed here as controls. Cells were lysed at different times during continuous culture and subjected to RNA isolation and analyzed by qPCR. Expression levels for the indicated genes are displayed relative to those in PANC-1 cells, set arbitrarily at 1.0. Data represent the means ± SD of 3–6 independent preparations. Please note the extra scales for BON and NT-3 cells in the SSTR2, 5 and CHGA graphs, which indicate the orders of magnitude higher expression. Data shown are the mean ± SD of at least three independent experiments. The asterisks (∗) denote significant differences (two-tailed unpaired Student’s *t*-test). The asterisks above BON and NT-3 cells indicate a significant difference relative to PANC-1 cells.

**Figure 2 cells-13-02010-f002:**
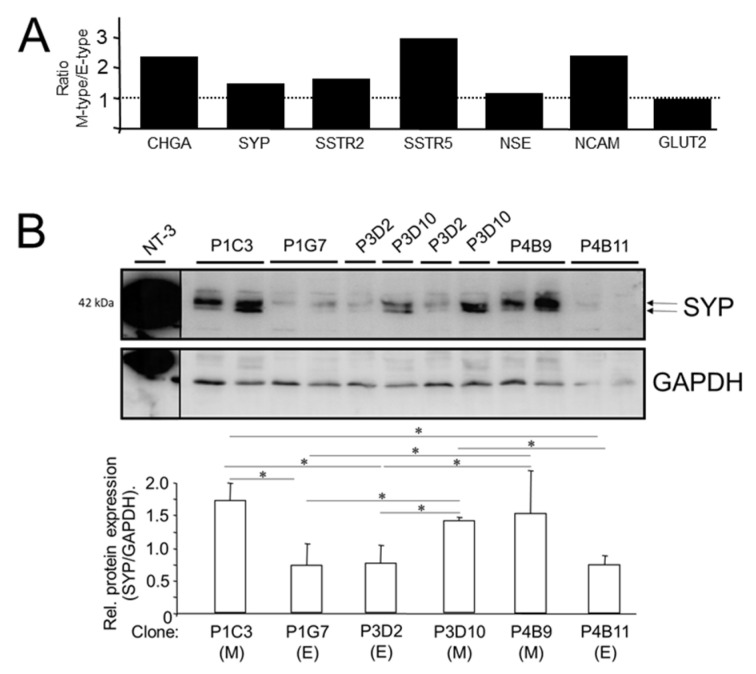
NED markers are enriched in PANC-1 single-cell-derived clones with an M-phenotype over those with an E-phenotype as indicated in parentheses in (**B**). (**A**) Seven individual clones previously grouped according to their EMT state (three E-type and four M-type clones) [6] were subjected to qPCR analysis for the indicated NED markers. Data represent the ratio of mean values between M and E-type clones for each marker. At 1.0, expression is equally high in M and E-type clones (indicated by the stippled line). (**B**) Immunoblot analysis of SYP in the same clones analyzed in (**A**). For quantitative analysis, the two closely spaced bands (indicated by arrows on the right-hand side) from two clones each (harvested at different times during continuous culture) were densitometrically scanned using the program NIH image. Hence, data represent the means ± SD of four bands per individual clone. Significant differences between the various clones are marked by asterisks (∗).

**Figure 3 cells-13-02010-f003:**
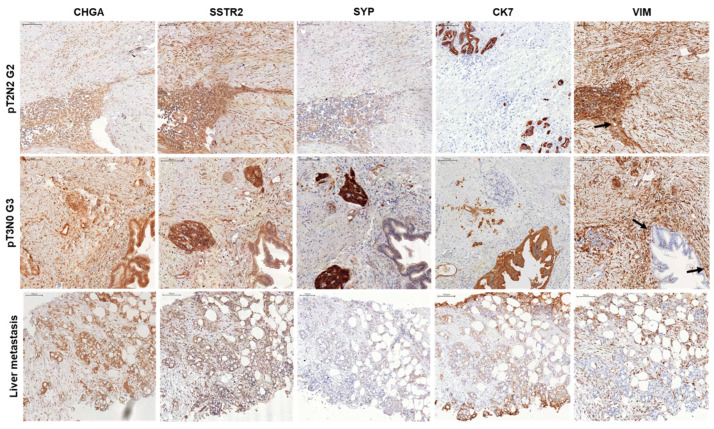
IHC of the indicated NED and EMT markers in human PDAC specimens, primary tumors and a liver metastasis. CHGA, SSTR2, and SYP are primarily, and CK7 exclusively, expressed in ductal epithelial tumor cells lining the pancreatic ducts, while VIM is present in tumor cells lining the pancreatic ducts (arrows) as well as in stromal fibroblasts (considered as an internal control). Representative images are shown. For negative controls, see Appendix A. The scalebar represents 100 µm.

**Figure 4 cells-13-02010-f004:**
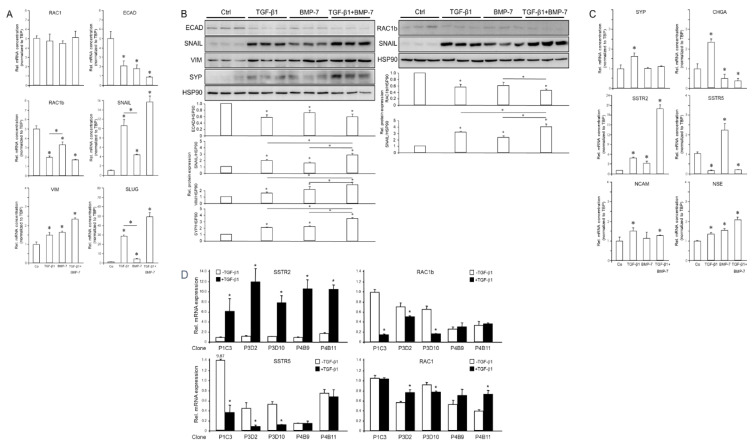
Effect of treatment with TGF-β1 or BMP-7 on EMT and NED markers in parental PANC-1 cells. Cells were treated with TGF-β1 (5 ng/mL), BMP-7 (200 ng/mL), or vehicle (Ctrl), either singly or in combination, for 24 h and subsequently subjected to qPCR (**A**) or immunoblot (**B**) analysis of the indicated EMT- and NED-associated genes. (**C**) As in (**A**), only NED-associated markers were detected. (**D**) Five single-cell-derived clones were treated with TGF-β1 (5 ng/mL) for 24 h and subjected to qPCR analysis for the indicated markers. The graphs below the blots in (**B**) display the results of densitometry-based quantification of band intensities. Data shown (mean ± SD of three parallel wells) are each from a representative experiment out of at least three experiments performed in total. The asterisks (∗) denote significant differences (two-tailed unpaired Student’s *t*-test).

**Figure 5 cells-13-02010-f005:**
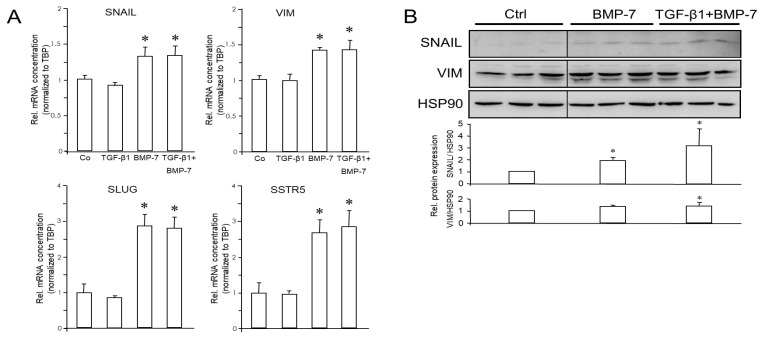
Effect of treatment with TGF-β1 or BMP-7 on EMT and NED markers in MIA PaCa-2 cells. Cells were treated with 5 ng/mL of TGF-β1, 200 ng/mL of BMP-7, or vehicle (Ctrl) either singly, or with both growth factors simultaneously, for 24 h and subjected to qPCR (**A**) or immunoblot (**B**) analysis of the indicated EMT- and NED-associated markers. The graphs below the blots in (**B**) display the results of densitometric band quantification. Data shown (mean ± SD of three parallel wells) are from a representative experiment out of at least three experiments performed in total. The asterisks (∗) denote significant differences (two-tailed unpaired Student’s *t*-test).

**Figure 6 cells-13-02010-f006:**
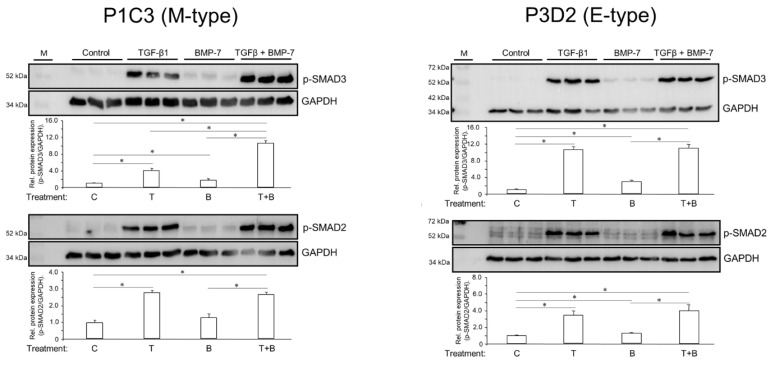
Effect of treatment with TGF-β1 or BMP-7 on the activation of SMAD3 and SMAD2 in PANC-1 clones P1C3 (M-type) and P3D2 (E-type). Cells (3 wells each) were treated for 1 h with either vehicle (Control, C), TGF-β1 (T, 10 ng/mL), BMP-7 (B, 200 ng/mL), either singly or in combination (TGF-β1+BMP-7, T+B), lysed, and subjected to phospho-immunoblotting of SMAD3 (upper panels) and SMAD2 (lower panels). The graphs below the blots display the results of densitometric signal quantification. Data shown (mean ± SD of three parallel wells) are from a representative experiment out of three experiments performed in total. The asterisks (∗) denote significant differences (two-tailed unpaired Student’s *t*-test).

**Figure 7 cells-13-02010-f007:**
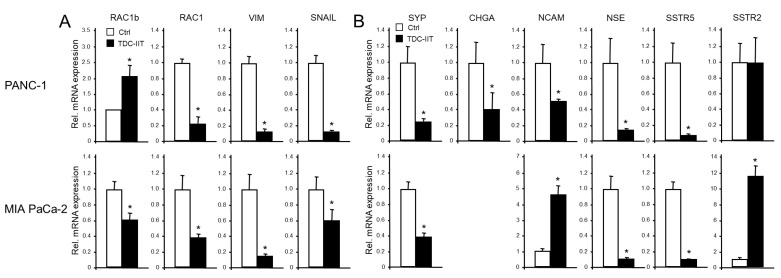
Expression of EMT and NED markers in response to MET induction via TDC-IIT. Parental PANC-1 (upper graphs) and MIA PaCa-2 (lower graphs) cells were subjected to TDC-IIT or control culture (Ctrl) for 72 h followed by lysis and qPCR analysis of markers of EMT (**A**) or NED (**B**). The data represent the mean ± SD of quadruplicate wells of a representative assay out of at least three assays performed in total. The asterisks (∗) denote significant differences relative to Ctrl.

**Figure 8 cells-13-02010-f008:**
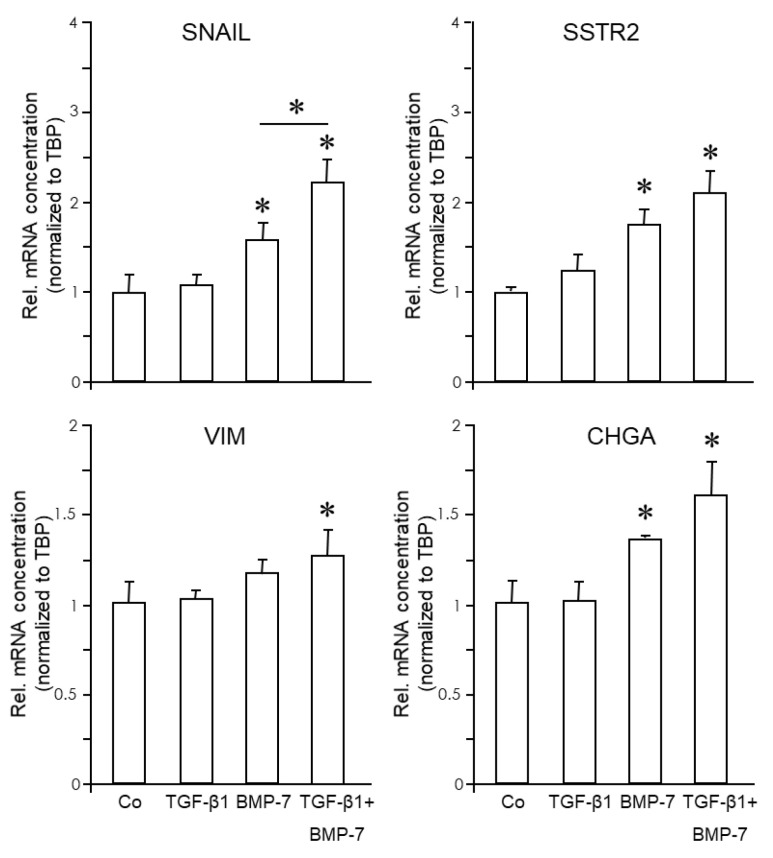
Expression of EMT and NED markers in response to TGF-β1 and BMP-7 in SCCOHT. SCCOHT-1 cells were treated with TGF-β1 (10 ng/mL) or BMP-7 (200 ng/mL) followed by lysis and qPCR analysis of markers of EMT (A) or NED (B). The graphs represent the mean ± SD of quadruplicate wells of a representative assay out of at least three independent assays performed in total. The asterisks (∗) denote a significant difference.

**Figure 9 cells-13-02010-f009:**
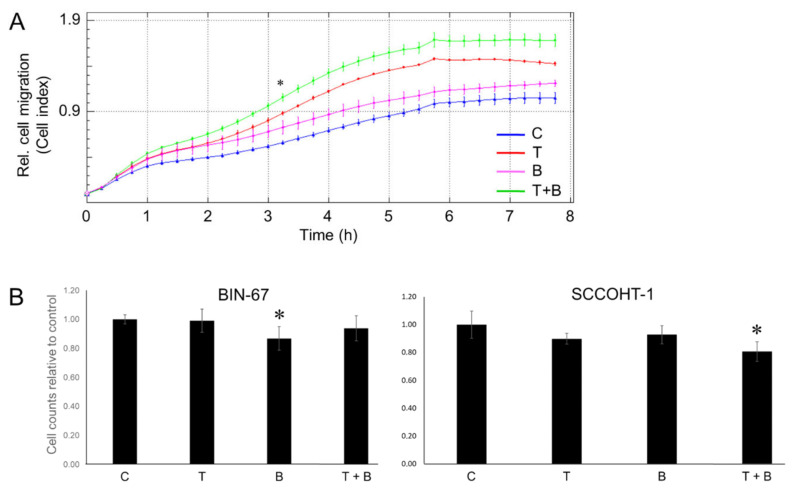
Impact of single and combined treatment with TGF-β1 and BMP-7 on functional activities in parental PANC-1, BIN-67, and SCCOHT-1 cells. (**A**) Migratory activity of parental PANC-1 cells as determined by RTCA using xCELLigence technology. PANC-1 cells received either vehicle (C) or were exposed to TGF-β1 (T, 10 ng/mL), BMP-7 (B, 200 ng/mL), or a combination of both growth factors (T+B) and ran on an xCELLigence device with CIM-plates-16 for a total of 7:45 h. Data were recorded by RTCA software and represent the mean ± SD of three parallel wells. A significant difference between the red and the green curves was first noted at the 3:25 h time point (indicated by the asterisk) and all later time points. (**B**) BIN-67 and SCCOHT-1 cells were treated for 24 h with either vehicle (C), TGF-β1 (T, 10 ng/mL), BMP-7 (B, 200 ng/mL), or a combination of both growth factors (T+B), after which cells were detached and counted. The graphs represent the mean ± SD of six independent assays (*n* = 6). The asterisks (∗) denote significant differences relative to C set arbitrarily to 1.00 (two-tailed unpaired Student’s *t*-test).

## Data Availability

The data that support the findings of the study are available from the corresponding author upon reasonable request.

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
