# Peer review of "Characterization of Epithelial–Mesenchymal and Neuroendocrine Differentiation States in Pancreatic and Small Cell Ovarian Tumor Cells and Their Modulation by TGF-β1 and BMP-7"

_cells, 2024, doi:10.3390/cells13232010_

Round 1
Reviewer 1 Report
Comments and Suggestions for Authors
The manuscript authored by Dr. Ungefroren, et al described a neuroendocrine differentiation state in pancreatic carcinoma cell lines that may associate with EMT. They concluded that in PDAC-derived tumor cells, NED is closely linked to EMT and TGF-43 β signaling, which may have implications for the therapeutic use of TGF-β inhibitors in PDAC management. Generally, this manuscript provided an interesting finding that may be useful in future clinical therapeutic strategy. However, the data provided are very preliminary data, not enough to draw solid conclusion. Most of results were only from qPCR, limited western blots are not convincing. No mechanistic data provided. I suggest the authors to add more data, using qPCR and western blotting in vitro, and IHC for tissues from different types of Pancreatic cancers. And need mechanistic data too.
Figure 1. add western blotting data to prove it. And add IHC from different type of PDAC tissues.
Figure 2. Need western blotting data too. If possible, perform immunofluorescence (IF) staining for these markers in the representative clones.
Figure 4. The western blots are not convincing.
Author Response
Dear Editor,
first of all, we would like to thank the reviewers for their enthusiastic comments on our manuscript. We have done our best to satisfy as many of their points of critique as possible and believe that this has considerably enhanced the overall quality of this work. All changes and additions to the original text have been highlighted in red color. Please note that due to the addition of a new figures to the original version, figure numbers in the main text and in the Supplementary material of the revised version have changed. Please also note that in response to a request of Reviewer 3 the title has been modified. In addition, we have removed duplications and have added a Conclusions section.
Reviewer 1
The manuscript authored by Dr. Ungefroren, et al described a neuroendocrine differentiation state in pancreatic carcinoma cell lines that may associate with EMT. They concluded that in PDAC-derived tumor cells, NED is closely linked to EMT and TGF-β signaling, which may have implications for the therapeutic use of TGF-β inhibitors in PDAC management. Generally, this manuscript provided an interesting finding that may be useful in future clinical therapeutic strategy. However, the data provided are very preliminary data, not enough to draw solid conclusion. Most of results were only from qPCR, limited western blots are not convincing. No mechanistic data provided. I suggest the authors to add more data, using qPCR and western blotting in vitro, and IHC for tissues from different types of Pancreatic cancers. And need mechanistic data too.
- Figure 1. add western blotting data to prove it. And add IHC from different type of PDAC tissues.
Response: We thank the reviewer for this suggestion. We presented new immunoblot analyses for SYP and SSTR2, while the CHGA antibody with this technique appeared at the detection limit likely due to the overall low-level expression of CHGA in the PDAC-derived cell lines (which is also evident from the high Ct values in qPCR analysis). SSTR2 protein was detectable in PANC-1 cells by immunoblotting with a decline in protein levels during MET induction (see Supplementary Figure 6, formerly Figure S2). Protein expression of SYP in PANC-1 cells and PDAC tissues is shown in Figures 2, 3 and 4.
As requested, the presence of CHGA, SSTR2 and SYP proteins was also demonstrated now by immunohistochemistry in different types of surgically resected tissue specimens of PDAC patients, including a liver metastasis. Specific staining was primarily localized to the ductal epithelial cells lining the pancreatic ducts. Please see new Figure 3 in the main text of the revised version and Supplementary Figure 3 for the corresponding negative controls.
- Figure 2. Need western blotting data too. If possible, perform immunofluorescence (IF) staining for these markers in the representative clones.
Response: As requested, we performed western blotting for SYP and SSTR2 in the PANC-1 clones. The SYP immunoblot has been included in Figure 2 as panel B. Unfortunately, IF staining was not possible since our antibodies did not work well in this application.
- Figure 4. The western blots are not convincing.
Response: We agree with the reviewer that the quality of this western blot could be improved. However, this is due to the low signal strength, which in turn is the result of the low physiological expression of SNAIL in MIA PaCa-2 cells. Nevertheless, by quantitatively analyzing several (underexposed) blots, we are confident to have accurately assessed the expression level of this EMT marker in MIA PaCa-2 cells.
- And need mechanistic data too.
Response: Mechanistic data (cell migration in PANC-1 cells and proliferation in BIN-67 and SCCOHT-1 cells) were also added to the revised version and are contained in the new Figure 9. Specifically, we observed additive or even synergistic effects of combined TGF-β1/BMP-7 treatment on the migratory activity of PANC-1 cells and on the proliferative activity of SCCOHT-1 cells (see also comment #3 of Reviewer 2 and comment #8 of Reviewer 3).
Reviewer 2 Report
Comments and Suggestions for Authors
The authors probe correlation of EMT and NED program by TGF beta in PDACs. Conventionally, BMP7 opposes TGF beta signaling in various contexts as is acknowledged by the authors in the text as well. However, in this context the authors observed and show data with additive effects of these growth factors on mRNA expression of EMT/MET and NED genes. To support the findings, it is essential to critically analyze the signaling activation by these GFs in the clones analyzed. Levels of pSMAD2/3/7 and pSMAD1/5 should be looked at alone in combinatorial treatments. Also, TGF beta signaling inhibitors may be utilized to assess that the effects seen are indeed driven by TGFbeta receptor activation. In addition to details on signaling, functional biological assays are required to truly appreciate the significance of the positive correlation of EMT/NED program in the clones.
Author Response
Dear Editor,
first of all, we would like to thank the reviewers for their enthusiastic comments on our manuscript. We have done our best to satisfy as many of their points of critique as possible and believe that this has considerably enhanced the overall quality of this work. All changes and additions to the original text have been highlighted in red color. Please note that due to the addition of a new figures to the original version, figure numbers in the main text and in the Supplementary material of the revised version have changed. Please also note that in response to a request of Reviewer 3 the title has been modified. In addition, we have removed duplications and have added a Conclusions section.
Reviewer 2
The authors probe correlation of EMT and NED program by TGF beta in PDACs. Conventionally, BMP7 opposes TGF beta signaling in various contexts as is acknowledged by the authors in the text as well. However, in this context the authors observed and show data with additive effects of these growth factors on mRNA expression of EMT/MET and NED genes. To support the findings, it is essential to critically analyze the signaling activation by these GFs in the clones analyzed. Levels of pSMAD2/3/7 and pSMAD1/5 should be looked at alone in combinatorial treatments. Also, TGF beta signaling inhibitors may be utilized to assess that the effects seen are indeed driven by TGFbeta receptor activation. In addition to details on signaling, functional biological assays are required to truly appreciate the significance of the positive correlation of EMT/NED program in the clones.
- critically analyze the signaling activation by these GFs in the clones analyzed. Levels of pSMAD2/3/7 and pSMAD1/5 should be looked at alone in combinatorial treatments
Response: As suggested, we have performed phospho-immunoblotting to assess the activation states of SMAD3 and SMAD2 in response to single and combined treatment with TGF-β1 and BMP-7. These data are shown in the new Figure 6 in the revised version. By comparing two subclones of PANC-1, one M- and one E-type clone, we observed that i) both growth factors were able to activate SMAD3 and SMAD2 and ii) only in the M-type clone TGF-β1 and BMP-7 acted in a synergistic manner to drive C-terminal phosphorylation and hence activation of SMAD3. Interestingly, SMAD3 is more important than SMAD2 in transcriptional regulation by TGF-β since in contrast to SMAD2, SMAD3 can bind directly to the Smad-binding-elements (SBEs) in the promoters of TGF-β responsive genes. We are currently analyzing additional M- and E-type clones to test if this differential synergistic activation of SMAD3 by the combined action of both growth factors is a true distinguishing feature.
- TGF beta signaling inhibitors may be utilized to assess that the effects seen are indeed driven by TGFbeta receptor activation:
Response: We used the ALK4/5/7 inhibitor, SB431542, during a 24-hour treatment of parental PANC-1 cells with TGF-β1. Using qPCR analysis of SNAIL, we found that this drug completely abolished the strong induction of SNAI1 by TGF-β1. In the same experimental setting we treated PANC-1 cells with BMP-7 in the absence or presence of the BMP/ALK2 inhibitor LDN193189. Like SB431542, this drug completely reversed the strong BMP-7 effect on SNAI1.These results, which are presented in Supplementary Figure 4 in the revised version, clearly show that the inducive effects of TGF-β1 and BMP-7 on EMT-associated genes are mediated by type I receptor activation, specifically ALK5 in the case of TGF-β1 and ALK2 in the case of BMP-7.
- In addition to details on signaling, functional biological assays are required to truly appreciate the significance of the positive correlation of EMT/NED program in the clones.
Response: As inquired, we have performed both migration and proliferation assays. Again, we observed additive or even synergistic effects of combined TGF-β1/BMP-7 treatment on the migratory activity of PANC-1 cells and on the proliferative activity of SCCOHT-1 cells (see new Figure 9 in the revised version). This issue was also raised by Reviewer 1 in his comment #4 and Reviewer 3 in his comment #8.
Reviewer 3 Report
Comments and Suggestions for Authors
This article provides new insights into EMT and NED in pancreatic ductal adenocarcinoma, and explores the roles of TGF-β1 and BMP-7 in regulating these processes. Overall, this is a very interesting work,but lacks some more in-depth mechanistic studies. Here are some suggestions:
1. Suggest to streamline the title, so that it is clear and direct.
2. The conclusions of the article are based on in vitro experiments, and STR identification of these cell lines is necessary to ensure the reliability of the conclusions.
3. In Figure 1, the expression of SSTR2 is missing data for BxPC-3 cell.
4. The section number for the "result" is incorrect.
5. In Section 3.2, the effects of TGF-β1 and BMP-7 on PANC-1 cells are described, but the text does not thoroughly discuss why BMP-7 promotes EMT, which is contrary to its known role in promoting MET in other cell types. The authors are advised to provide a possible explanation or further experiments to explore this phenomenon. Additionally, does the stimulation of TGF-β1 and BMP-7 yield the same results in moderately differentiated pancreatic cancer cells BxPC-3 and immortalized pancreatic duct epithelial cells HPDEH6c7?
6. Considering that BMP-7 shows synergistic effects with TGF-β1 in some cases, but antagonistic effects in others, it is recommended to conduct more in-depth mechanistic studies to determine how these factors affect EMT and NED through different signaling pathways.
7.The article primarily focuses on changes in gene expression levels. It is suggested that additional functional experiments be conducted to verify the impact of these gene expression changes on cellular behaviors, such as cell migration, invasion, or proliferation assays.
8. The article utilized specific concentrations of TGF-β1 and BMP-7. It is suggested to explore the effects of these factors on cells at varying concentrations to determine whether there are dose-dependent effects.
9. It is recommended to use clinical samples (such as TCGA) to validate the EMT and NED-related gene expression patterns observed in cell lines. It is also suggested to perform immunohistochemical analysis on tissue microarrays to examine changes in protein levels (if possible), in order to enhance the clinical relevance of the study's findings.
Author Response
Dear Editor,
first of all, we would like to thank the reviewers for their enthusiastic comments on our manuscript. We have done our best to satisfy as many of their points of critique as possible and believe that this has considerably enhanced the overall quality of this work. All changes and additions to the original text have been highlighted in red color. Please note that due to the addition of a new figures to the original version, figure numbers in the main text and in the Supplementary material of the revised version have changed. Please also note that in response to a request of Reviewer 3 the title has been modified. In addition, we have removed duplications and have added a Conclusions section.
Reviewer 3
This article provides new insights into EMT and NED in pancreatic ductal adenocarcinoma, and explores the roles of TGF-β1 and BMP-7 in regulating these processes. Overall, this is a very interesting work,but lacks some more in-depth mechanistic studies. Here are some suggestions:
- Suggest to streamline the title, so that it is clear and direct.
Response: As requested, the title has been streamlined to enhance clarity.
- The conclusions of the article are based on in vitro experiments, and STR identification of these cell lines is necessary to ensure the reliability of the conclusions.
Response: As requested, all cell lines used in this study have been STR-typed. The results of the STR fragment analysis confirmed their identity and are shown in the new Supplementary Figure 1. The former Figure S1 has become Supplementary Figure 2.
- In Figure 1, the expression of SSTR2 is missing data for BxPC-3 cell.
Response: The expression data of SSTR2 for the BxPC-3 cell line have been added.
- The section number for the "result" is incorrect.
Response: The section numbers for the “Results” section (3.1. through 3.9.) have been corrected.
- In Section 3.2, the effects of TGF-β1 and BMP-7 on PANC-1 cells are described, but the text does not thoroughly discuss why BMP-7 promotes EMT, which is contrary to its known role in promoting MET in other cell types. The authors are advised to provide a possible explanation or further experiments to explore this phenomenon.
Response: As requested, we have carried out additional experiments to explore this phenomenon (see also comment #1 of Reviewer 2): We have performed phospho-immunoblotting to assess the activation states of SMAD3 and SMAD2 in response to single and combined treatment with TGF-β1 and BMP-7. These data are shown in the new Figure 6 in the revised version. By comparing two subclones of PANC-1, one M-type and one E-type clone, we observed that i) both growth factors were able to activate SMAD3 and SMAD2 and ii) only in the M-type clone TGF-β1 and BMP-7 acted in a synergistic manner to drive C-terminal phosphorylation and hence activation of SMAD3. Interestingly, SMAD3 is more important than SMAD2 in transcriptional regulation by TGF-β since in contrast to SMAD2, SMAD3 can bind directly to the Smad-binding-elements (SBEs) in the promoters of TGF-β responsive genes. We are currently analyzing additional M- and E-type clones to test if this differential synergistic activation of SMAD3 by the combined action of both growth factors is a true distinguishing feature. With these data we have provided a possible explanation of why BMP-7 promotes EMT. Our results are in line with data of another study that also employed PANC-1 cells (new Ref. 45 in the revised version).
- Additionally, does the stimulation of TGF-β1 and BMP-7 yield the same results in moderately differentiated pancreatic cancer cells BxPC-3 and immortalized pancreatic duct epithelial cells HPDEH6c7?
Response: We are grateful for this point. Accordingly, we have tested by qPCR analysis if the stimulation of BxPC-3 and HPDEH6c7 cells with TGF-β1 and BMP-7 yields the same results as in PANC-1 cells. We found that both cell types responded only moderately or not at all to both growth factors with respect to EMT (ECAD, SNAIL, SLUG, VIM) and NED (SSTR2, SSTR5) markers and rarely exhibited additive or synergistic effects on simultaneous treatment with both growth factors. These new data are contained in the new Supplementary Figures 5 (BxPC-3) and 6 (HPDE) in the revised version.
- Considering that BMP-7 shows synergistic effects with TGF-β1 in some cases, but antagonistic effects in others, it is recommended to conduct more in-depth mechanistic studies to determine how these factors affect EMT and NED through different signaling pathways.
Response: This issue was also raised by Reviewer 2 in his comment #1. We have monitored the activation states of the Smad proteins involved in signaling by TGF-β1 (p-SMAD2 and p-SMAD3). For details, see above (our response to your comment #5).
- The article primarily focuses on changes in gene expression levels. It is suggested that additional functional experiments be conducted to verify the impact of these gene expression changes on cellular behaviors, such as cell migration, invasion, or proliferation assays.
Response: As requested, we have performed both migration and proliferation assays. Again, we observed additive or even synergistic effects of combined TGF-β1/BMP-7 treatment on the migratory activity of PANC-1 cells and on the proliferative activity of SCCOHT-1 cells (see new Figure 9 in the revised version). This issue was also raised by Reviewer 1 in his comment #4.
- The article utilized specific concentrations of TGF-β1 and BMP-7. It is suggested to explore the effects of these factors on cells at varying concentrations to determine whether there are dose-dependent effects.
Response: We have evaluated the effect of varying concentrations of TGF-β1 and BMP-7 on the expression of the common target gene SNAI1 (encoding SNAIL). Results show that both factors induce this gene in dose-dependent manner. These data are shown in the new Supplementary Figure 4.
- It is recommended to use clinical samples (such as TCGA) to validate the EMT and NED-related gene expression patterns observed in cell lines. It is also suggested to perform immunohistochemical analysis on tissue microarrays to examine changes in protein levels (if possible), in order to enhance the clinical relevance of the study's findings.
Response: Thank you for this point. As suggested, we have performed immunohistochemical analysis on tissue specimens of surgically resected tumor tissues from PDAC patients and have successfully demonstrated the presence of CHGA, SSTR2 and SYP proteins primarily in the ductal epithelial cells lining the pancreatic ducts. Please see new Figure 3 in the main text of the revised version and Supplementary Figure 3 for the corresponding negative controls. See also comment # 1 of Reviewer 1.
Round 2
Reviewer 1 Report
Comments and Suggestions for Authors
The author has made effort to make the manuscript much better. To further improve the manuscript and make the conclusion solid, the following suggestions need to be addressed.
Figure 1. Add statistical analysis and p-value. Why only showing the western for BxPC3 and NT3 in supplementary fig.2? Need to show protein levels of each protein for all cell lines together, even some of them are undetectable for comparison.
Figure 2. Please label which subclones are M-type and which are E-type.
Figure 3. It is better to stain some epithelial markers such as CK and mesenchymal markers such as vimentin in the same tissues and analyze if there is any correlation between these markers and NED markers.
Author Response
We would like to thank reviewer 1 for his appreciation of our revisions. We have done our best to satisfy the additional points of his critique. All changes and additions to the original text have been highlighted again in red color. The reconsideration of our work is appreciated. Specifically, we would like to respond to the critique of Reviewer 1 as follows:
The author has made effort to make the manuscript much better. To further improve the manuscript and make the conclusion solid, the following suggestions need to be addressed.
Figure 1. Add statistical analysis and p-value. Why only showing the western for BxPC3 and NT3 in supplementary fig.2? Need to show protein levels of each protein for all cell lines together, even some of them are undetectable for comparison.
Response: As requested, we have added the statistical analysis with an indication in the figure of whether differences are significant (p < 0.05, denoted by asterisks).
We should mention that the focus of our study is on the mesenchymal-type cell lines, PANC-1 and MIA PaCa-2 cells. The neuroendocrine tumor cell lines, BON and NT-3 served as positive controls and for this reason were not included in all Western blots. For PANC-1 and MIA PaCa-2 cells, data on protein expression for all NED and EMT markers are contained in the manuscript, either as immunoblot and/or IHC data. Moreover, during the first revision we had included more expression (qPCR) data for BxPC3 and HPDE cells in response to a request from reviewer 3 (see Figures S5 and S6).
Figure 2. Please label which subclones are M-type and which are E-type.
Response: As requested, we have indicated in panel B in parentheses the respective EMT subtypes underneath the clone names.
Figure 3. It is better to stain some epithelial markers such as CK and mesenchymal markers such as vimentin in the same tissues and analyze if there is any correlation between these markers and NED markers.
Response: This is a very good suggestion. We have therefore stained adjacent tissue sections with the epithelial marker CK7 and the mesenchymal marker vimentin. CK7 was chosen because its overexpression in cancer is associated with increased proliferation, migration, metastasis and TGF-β induced EMT (for reference see new Ref. #37). We observed a nice colocalization between these EMT markers, in particular CK7, and the NED markers. The staining for vimentin was also positive in most tumor cells lining the ducts and in stromal fibroblasts (for reference see new Ref. #38). These data have been added to Figure 3 on the right-hand side.
Reviewer 3 Report
Comments and Suggestions for Authors
My concerns about the manuscript have been resolved.
Author Response
My concerns about the manuscript have been resolved.
Response: We are delighted that reviewer 3 was happy with our revisions.